# Polyvinyl Alcohol Reinforced Flame-Retardant Polyacrylonitrile Composite Fiber Prepared by Boric Acid Cross-Linking and Phosphorylation

**DOI:** 10.3390/ma11122391

**Published:** 2018-11-27

**Authors:** Yuanlin Ren, Tian Tian, Lina Jiang, Xiaohui Liu, Zhenbang Han

**Affiliations:** 1School of Textiles, Tianjin Polytechnic University, Tianjin 300387, China; 2Key Laboratory of Advanced Textile Composite, Ministry of Education, Tianjin Polytechnic University, Tianjin 300387, China; tiantian@163.com (T.T.); jianglina@163.com (L.J.); hanzhenbang@163.com (Z.H.); 3School of Materials Science and Engineering, Tianjin Polytechnic University, Tianjin 300387, China; liuxiaohui@163.com

**Keywords:** polyacrylonitrile, poly (vinyl alcohol), flame retardance, high strength, composite fiber, phosphorylation

## Abstract

To improve the strength and maintain the inherent properties of flame-retardant polyacrylonitrile (FR-PAN) fiber, a commercialized hydrocarbon polymer, i.e., poly (vinyl alcohol) (PVA), used as an enhancement component, was blended with polyacrylonitrile (PAN) spinning dope to fabricate a PVA/PAN composite fiber through wet-spun technology. Then, cross-linked PVA/PAN composite fiber (C-PVA/PAN) was acquired via boric acid cross-linking. Finally, flame-retardant C-PVA/PAN fiber (FR-PVA/PAN) was prepared by phosphorylation. The structures of the samples were characterized by Fourier transform infrared analysis (FTIR) and X-ray photoelectron spectroscopy (XPS). The thermogravimetric analysis (TGA) results reveal that the thermal stability of the composite fiber is lower than that of the pristine PAN fiber. However, the char residue of the composite fiber is higher than that of the control sample, wherein, FR-PVA/PAN has the highest char residue of 62.5 wt% at 800 °C. The results regarding the combustion properties of FR-PVA/PAN show that the fire hazard of FR-PVA/PAN is restrained greatly, indicating excellent flame-retardant performance. The corresponding flame-retardant mechanism of FR-PAV/PAN is investigated by Pyrolysis gas chromatography and mass spectrometry (Py-GC/MS) and thermogravimetric analysis coupled with Fourier transform infrared analysis (TG-FTIR). The results indicate the gas-phase and condensed-phase flame-retardant mechanisms.

## 1. Introduction

Polyacrylonitrile (PAN) fiber is one of the most important synthetic fibers in the textile field. It is spun from polyacrylonitrile, particularly the acrylonitrile (AN) copolymer with more than 85% acrylonitrile contents. PAN fiber has many excellent properties, such as good elasticity, soft handle, excellent warmth retention, and good dyeing properties. Therefore, PAN fiber has gained a good reputation for being an “artificial wool”. However, the limiting oxygen index (LOI) value of PAN fiber is around 17%, which is one of the most combustible synthetic fibers. In addition, PAN fiber will produce toxic hydrogen cyanide and heavy smoke once on fire. Hence, it becomes particularly important to impart PAN fiber with flame-retardant properties to extend their application.

Flame-retardant PAN or its fiber can be produced by different kinds of methods, such as copolymerization, surface modification, or blending. The copolymerization technique can introduce flame-retardant elements into the macromolecular chain of an acrylonitrile copolymer via the covalent bond, resulting in durable flame-retardant performance [1,2,3]. In this scenario, those monomers containing flame-retardant elements, especially phosphorus-containing comonomers, must have previously undergone molecular design. The cost will rise to some extent, in addition, the comonomers applied for copolymerization with acrylonitrile have a relatively high molecular weight and huge bulk volume. As a result, the copolymerization becomes difficult to operate due to steric effects, and the high molecular weight of acrylonitrile copolymer is arduous to synthesize. Therefore, the produced PAN fiber will have poor mechanical properties. Hence, some commercialized esters and acids, such as methacrylate [4], methacrylic acid [5], and itaconic acid [5,6,7] are used to copolymerize with acrylonitrile. The inherent or released acid groups of these copolymers improve the cyclization reaction of the cyano group of the acrylonitrile copolymer upon heating, resulting in an increase in the char-forming ability and the suppression of further decomposition. Unfortunately, the flame-retardant ability of these kinds of copolymers is far less than those of acrylonitrile copolymers containing phosphorus elements. 

Surface modification is an easy and cost-effective technique to confer PAN fiber flame retardancy. Polyamine, such as hydrazine hydrate [8,9] and diethylenetriamine [10] were reacted, respectively, with PAN fiber through nitrile groups and amine groups. In this way, durable flame-retardant PAN fiber can be obtained. However, the mechanical performance of this kind of fiber is poor. Moreover, the yellow or brown color of the fiber restricts its broad application. To overcome the mentioned drawbacks, we used hydroxylamine hydrochloride to react with a PAN fabric, followed by phosphorylation to obtain a durable flame-retardant PAN fabric [11]. The LOI value is 29.8% after 20 cycles of washing, which shows a good durable flame-retardant performance. The color of the modified PAN fabric is almost unchanged; however, the fabric becomes stiff to some extent.

Nowadays, the sol-gel technique has become a popular solution to prepare flame-retardant materials. In our previous work, tetraethoxysilane (TEOS) was used as the silicane precursor to prepare silica sol. PAN fabric was dipped into the silicane sol containing polyphosphoric acid [12] or phytic acid and urea [13], followed by gelation to prepare the flame-retardant PAN fabric. Furthermore, a cationic silicon hydrogel which was prepared by using 3-aminopropyltriethoxysilane (KH550) and an anionic solution of phytic acid was utilized to treat the PAN fabric via a sol-gel and a layer-by-layer process. The flame-retardant performance of the PAN fabric improved greatly compared to the control sample. Similarly, the fabric treated with a sol-gel process or a sol-gel and layer-by-layer process became stiff, and the application was restricted accordingly.

Flame-retardant PAN fiber can be gained through the blending technique. In such a scenario, most importantly, flame retardants must be dissolved or uniformly dispersed in PAN spinning dope. Then, flame-retardant PAN fiber can be fabricated via the wet-spun process. Flame retardants may be lost during the spinning or washing process. In order to obtain the required flame-retardant property, many flame retardants (over 25 wt%) should be added. As a result, the spinning is more difficult and the breaking strength of the fiber decreases. In line with this reason, under the premise of ensuring flame retardancy, it is of great practical significance to improve the physical properties of the PAN fiber. In order to improve the mechanical properties of flame-retardant polyacrylonitrile fiber, Zhou [14] blended 30–50 wt% of polyvinyl alcohol (PVA) with a polyacrylonitrile spinning solution to fabricate blended PAN fiber followed by thermal cross-linking. Then, the fiber was treated with hydrazine hydrate and NaOH aqueous solution in turn. The mechanical and flame-retardant properties increased. However, the polyacrylonitrile-based fiber treated with a polyamine, such as hydrazine hydrate [8,9] and diethylenetriamine [10] implies that the cyano group of the polyacrylonitrile fiber changes into other nitrogen-containing groups. In other words, the increased flame-retardant performance is at the expense of the inherent properties of the pristine PAN fiber, owing to the loss of a large number of cyano groups.

Herein, we explored an improved blending modification technique to prepare flame-retardant polyvinyl alcohol/polyacrylonitrile composite fiber (FR-PVA/PAN). Firstly, a given mass of PAN and PVA was dissolved in dimethyl sulfoxide (DMSO) and wet-spun to fabricate PVA/PAN composite fiber. Then, the composite fiber was delivered into a boric acid coagulation solution with a concentration of about 5 wt% and drawn to obtain cross-linked PVA/PAN composite fiber (C-PVA/PAN). After that, the mentioned fiber was phosphorylated with phosphorus acid and catalyzed by urea to prepare the flame-retardant PVA/PAN composite fiber (FR-PVA/PAN). The structure, flame-retardant properties, and flame-retardant mechanism were investigated in detail.

## 2. Experimental Section

### 2.1. Materials

Polyacrylonitrile (PAN) powder (95 wt% acrylonitrile and 5 wt% vinyl acetate) was supplied by Jilin Chemical Fiber Co., Jilin, China. Dimethylsulfoxide (DMSO), polyvinyl alcohol (PVA) (average molecular weight: 65,000; polymerization degree: 1700 ± 50; hydrolysis degree: ca. 80%), and boric acid (BA) were purchased from Tianjin Kermal Chemical Reagent Co., Ltd., Tianjin, China. Phosphorus acid (85 wt%) and urea were obtained from Tianjin Fengchuan Chemical Reagent Co., Ltd., Tianjin, China. All the reagents were analytical grade and were used without any further purification.

### 2.2. Preparation of Cross-Linked PVA/PAN Composite Fiber (C-PVA/PAN)

In a 250 mL three-necked round-bottom flask equipped with a mechanical agitator, 8.5 g of DMSO was added and stirred at ambient temperature. A total of 1.2 g of PAN and 0.3 g of PVA were added to the flask in batches. Then, the mixture was gradually heated to 80 °C for 1.5 h. After that, the uniform PVA/PAN spinning dope was statically defoamed for 24 h. The deaerated PVA/PAN spinning dope was transferred to a spinning pump with a nitrogen inlet. The spinning dope was extruded from a spinneret orifice (0.05 mm) into a boric acid aqueous solution with the concentration of about 5 wt% at 80 °C. After that, the cross-linked PVA/PAN composite fiber (C-PVA/PAN) was continuously drawn in hot water and dry air to obtain highly oriented composite fiber. Finally, the fiber was dried in a vacuum oven at 60 °C for 6 h to a constant weight. The control PAN fiber and PVA/PAN composite fiber were prepared in the same way except for the cross-linking.

### 2.3. Preparation of Flame Retardant PVA/PAN Composite Fiber (FR-PVA/PAN)

In a 500 mL beaker containing 300 mL of 50 wt% phosphoric acid aqueous solution, 5 g of urea was added and stirred at 80 °C until the urea was completely dissolved. C-PVA/PAN (1 g) was immersed in the mentioned solution for 1 h. After that, the flame-retardant PVA/PAN composite fiber (FR-PVA/PAN) was taken out, washed with deionized water many times, and dried in a thermostatic oven at 60 °C to a constant weight. The schematic route is illustrated in Scheme 1.

### 2.4. Characterization

Fourier transform infrared spectroscopy (FTIR) of the fabrics was performed on a Bruker Vector 22 spectrometer (Bruker, Billerica, MA, USA). The spectra ranged from 400 to 4000 cm^−1^ were recorded with a resolution ratio of 4 cm^−1^.

The X-ray photoelectron spectroscopy (XPS) was utilized to analyze the element composition using an X-ray photoelectron spectrometer (Thermofisher, Wsltham, MA, USA) with Al Kα excitation radiation (hν = 1486.6 eV).

Thermogravimetric (TG) analysis was carried out on a thermogravimetric analyzer (Thermal Analysis Q600 SDT, New Castle, DE, USA) to study the thermal stability. All samples were heated from room temperature to 800 °C at a heating rate of 10 °C/min under an air flow of 30 mL/min.

Differential scanning calorimetry (DSC) was performed on a DSC200F3 (Netzsch, Ahlden, Germany). A precise weighing sample (5 mg) was sealed in aluminum sample pans. The samples were scanned under N_2_ (30 mL/min) from room temperature to 350 °C at a heating rate of 10 °C/min. DSC curves of all the samples were obtained from a single heating process.

The fiber linear density and mechanical properties of the samples were measured through an automatic single-fiber universal physical property tester (FAVIMAT-AIROBOT, Textechno, Germany) with a 2.5 cN pretension force according to DIN53816. The samples were tested with a gauge length of 20 mm and crosshead velocity of 2 mm/min at a strain rate of 10%/min. The testing was performed at a constant temperature of 25 ± 1 °C and a relative humidity of 65 ± 1%. A total of 25 fibers for every fiber sample were measured and averaged.

The fabrics of different fibers were woven with the basic weight of 165 ± 5 g/m^2^. The HC-2 limited oxygen index instrument was used to test the limiting oxygen index (LOI) of the samples according to GB 5454-85. Samples of size 5 mm × 15 mm were ignited for several seconds in a methane flame and tested five times.

The flame-retardant durability of FR-PVA/PAN fabric was tested according to an AATCC Test Method 61-2003 by using 0.37 wt% detergent. The size of the fabrics was 5 cm × 10 cm. One washing cycle lasted 45 min, which equals five commercial launderings.

The combustion performance of PAN and FR-PVA/PAN fabrics with the size of 100 mm × 100 mm × 2 mm was performed on a cone calorimeter (FTT, East Grinstead, UK) according to ISO 5660-1. The irradiative heat flux in a horizontal configuration was 35 kW/m^2^. The parameters, such as time to ignite (TTI), heat release rate (HRR), peak of HRR (PHRR), total heat release (THR), smoke production rate (SPR), total smoke production (TSP), and the peak of SPR (PSPR) were evaluated. In addition, fire growth rate index (FIGRA) defined as the ratio of PHRR and time to PHRR, was obtained.

Thermogravimetric analysis coupled with Fourier transform infrared analysis (TG-FTIR) was carried out on a combined STA 6000 Frontier TGA (Perkin Elmer, Waltham, MA, USA) and Nicolet FTIR (Thermo Fisher Science, Waltham, MA, USA). The decomposed gases of the sample were transferred from the TGA analyzer to the TG-FTIR interface through a stainless steel transfer pipe. The testing was performed under nitrogen with a flow rate of 50 mL/min. The released gases were measured by an FTIR spectrometer with 2 cm^−1^ resolution and 500–4000 cm^−1^ scanning range.

The pyrolysis gaseous products of the FR-PVA/PAN fabric were measured by pyrolysis–gas chromatography/mass spectrometry (Py-GC/MS). Py-GC/MS was conducted on a pyroprobe (EGA/PY 3030D; Frontier, Tokyo, Japan) and a gas chromatography–mass spectrometry analyzer (6890N; Agilent, Santa Clara, CA, USA). The temperature of pyrolysis ranged from room temperature to 537 °C under a helium atmosphere at a heating rate of 10 °C/min. After pyrolysis, the volatile products were transferred to the gas chromatography (GC) injector with the setting temperature of 280 °C.

## 3. Results and Discussion

### 3.1. Mechanical Properties

The mechanical properties of the different samples are listed in Table 1. Under the same spinning conditions, the linear density of PVA/PAN is higher than that of the control fiber. It is generally known that PVA and PAN are two different macromolecules. They have different crystalline structures and motion ability. In addition, inter- and intra-hydrogen bonds between PVA and PAN macromolecules restrain the movement of the two macromolecules. Thus, under the same spinning conditions, the two macromolecular chains may be crimp and not completely drawn during stretching. Therefore, the linear density of the PVA/PAN composite fiber increases accordingly. For C-PVA/PAN, the linear density increases a little. The cross-linking reaction between boric acid and PVA changes the macromolecular orientation and the arrangement by generating a network, thus the linear density of the C-PVA/PAN composite fiber increases. However, the increase of the linear density of the FR-PVA/PAN fiber is slightly higher because the phosphorylation introduces phosphorus-containing groups on the surface of the composite fiber. The tensile strength of the samples, such as PVA/PAN composite fiber and the modified composite fiber, increases. This may be due to the presence of some interaction between the PAN and PVA macromolecules in the blend [14]. However, the tensile strength of FR-PVA/PAN decreases, attributed to a certain degree of degradation of the PAN chains caused by phosphoric acid modification. As far as the breaking elongation is concerned, the breaking elongation of the PVA/PAN composite fiber and the modified composite fiber decreases compared with the control PAN fiber. As previously mentioned, the PVA/PAN composite fiber and the modified composite fiber are highly oriented and stretched. Therefore, the deformation of the macromolecular chains of PVA and PAN is very small when the fibers are pulled off. In addition, the flame-retardant modification by phosphorus acid may cause structural damage to the fiber, decreasing the breaking elongation of the fibers [14].

### 3.2. FTIR Analysis

The chemical structures of the fibers were analyzed by FTIR, as shown in Figure 1. The original PAN fiber has five main absorption peaks. The absorption peak at 2935 cm^−1^ was due to the stretching vibration of the C–H bond in CH_2_, and the peak at 2241 cm^−1^ was ascribed to the stretching vibration of –CN. In addition, the absorption peaks at 1736 cm^−1^, 1451 cm^−1^, and 1236 cm^−1^ belonged to the stretching vibration of the C=O bond in the ester group of the vinyl acetate unit, the C–H bending vibration in CH_2_, and the bending vibration of C–N in CN [15]. For the PVA/PAN composite fiber, except for the characteristic peaks of PAN, the broad peak at approximately 3470 cm^−1^ was considered as the stretching vibration of OH in PVA, and the new emerged peak at 1630 cm^−1^ was mainly attributed to the stretching vibration of C=O of the acetate groups, which remained in the partially hydrolyzed PVA [16,17]. For the C-PVA/PAN composite fiber, new absorption peaks located at 772 cm^−1^ and 653 cm^−1^ were assigned to the stretching vibration of B–O–C and O–B–O [17,18]. The results showed that the chemical cross-linking reaction had successfully taken place between boric acid and PVA/PAN. In the case of FR-PVA/PAN, the peak at 1650 cm^−1^ was attributed to the carboxyl group (–COOH), which demonstrated that the CN groups in PAN were hydrolyzed continuously in the phosphoric acid solution. Furthermore, the newly appeared absorption peak at 1160 cm^−1^ was due to the stretching vibrations of P=O, P–O–C, and P–O. The stretching vibration absorption peak of P–O–N appeared at 973 cm^−1^ [15,19]. 

### 3.3. XPS Analysis

It is reported that acrylonitrile polymer has good flame-retardant performance when the phosphorus content in the polymer is over 3 wt% [1,5]. Therefore, it is necessary to know the content of each element in the fiber samples. The chemical compositions of PAN, PVA/PAN, C-PVA/PAN, and FR-PVA/PAN were analyzed by XPS, as shown in Figure 2 and Table 2. The four samples all contained C_1s_, N_1s_, and O_1s_, corresponding to the peaks at 300 eV, 409 eV, and 543 eV, respectively [11,12,13]. Compared with PAN fiber, the PVA/PAN fiber showed an increase of the oxygen element from 10.86% to 14.28%. This increase was mainly due to the hydroxyl groups of PVA. In the case of the C-PVA/PAN fiber, the peak at 191 eV was assigned to B_1s_ [20]. For FR-PVA/PAN, a new peak at 131 eV, assigned to P_2p_ [11,12,13], clearly appeared. Thus, it can be concluded that the chemical modification of C-PVA/PAN with phosphoric acid was successful. Furthermore, the phosphorus and boron contents are as much as 7.70 wt% and 13.18 wt%, respectively, which helps to improve the flame retardancy of FR-PVA/PAN. 

### 3.4. Thermal Stability

TG-DTG and DSC techniques are convenient and effective for the assessment of the thermal properties of materials. Figure 3 shows the TG (a) and DTG (b) curves of different fiber samples under an air atmosphere.

As shown in Figure 3b, three decomposition steps were observed on the TG curves of the original PAN fiber. The first stage took place from 265 °C to 347 °C. The maximum weight loss rate occurred at 290 °C due to a cyclization reaction that was accompanied by the loss of ammonia and hydrogen cyanide [1]. The second stage ranged from 347 °C to 470 °C with a maximum weight loss rate at 370 °C due to the decomposition and carbonization process. The third stage, which started from 470 °C to 779 °C corresponded to the thermo-oxidation process [13,21]. Furthermore, with the occurrence of oxidative cracking, the mass loss reached its maximum during the whole thermal degradation process. The maximum weight loss rate was observed at 672 °C.

For the PVA/PAN fiber, it exhibited five decomposition steps. The first stage ranged from 44 °C to 200 °C with the maximum weight loss rate observed at 152 °C. The weight loss was mainly due to the evaporation of the bound water [21]. The second stage was from 152 °C to 263 °C with the maximum weight loss rate observed at 239 °C due to the side chain degradation of PVA [17,22] and the cyclization of PAN. The third stage ranged from 263 °C to 354 °C with the maximum weight loss rate observed at 302 °C. The weight loss was ascribed to the main chain decomposition of PVA, its further degradation into char [23], and the subsequent cyclization and dehydrogenation of PAN. It was clearly found that the third stage’s initial decomposition temperature of the PVA/PAN fiber decreased significantly compared to that of the control PAN fiber. It indicated that the addition of PVA effectively promoted the cyclization of PAN. The fourth stage began from 354 °C to 477 °C with the maximum weight loss rate observed at 423 °C. The decomposition mechanism of this stage was similar to that of pure PAN. The decomposition and carbonization resulted in the formation of the trapezoidal structure as well as the further thermal-oxidative decomposition of the resulted char of PVA.

The C-PVA/PAN composite fiber possessed four thermal degradation stages. The first stage ranged from 128 °C to 282 °C, contributing to the degradation of PVA, as mentioned above. The second weight loss started from 282 °C to 403 °C with the maximum weight loss rate observed at 307 °C, which was mainly due to the cyclization of PAN and the main chain decomposition of PVA. Because of the occurrence of the decomposition and carbonization of PAN, the weight loss rate reached the maximum at 427 °C in the third stage. In the final stage, C-PVA/PAN furtherly underwent thermal-oxidative degradation, starting from 559 °C. Interestingly, FR-PVA/PAN had the highest carbon residue at 800 °C (ca. 62.5 wt%), which is higher than that of PAN (no residue left), PVA/PAN (ca. 3.1 wt%), and FR-PVA/PAN (ca. 5.1 wt%). The reason for this was that the introduction of the phosphorus group could be decomposed to a phosphorous-containing acid, which functions as nucleophilic centers to accelerate the cyclization of the cyano groups in the FR-PVA/PAN to form an intumescent char layer [24]. Furthermore, the addition of PVA also promotes the formation of a carbon layer in the cyclization process. Moreover, the chemical cross-linking reaction had successfully occurred between boric acid and PVA/PAN, and the cross-linked fiber possessed a network structure which could act as an excellent flame-retardant barrier.

The DSC curves of different fibers in nitrogen were shown in Figure 4 and the relevant data were listed in Table 3. The DSC tests were carried out under a nitrogen atmosphere; therefore, no oxidative reactions happen during the testing procedure. As shown in Figure 4, the control PAN fiber displayed one sharp and shoulder-containing exothermic peak at 306 °C, which was mainly due to the cyclization of PAN [25]. The cyclization reactions are initiated through a free-radical mechanism [26]. However, PVA/PAN exhibited an endothermic peak at 93 °C, ascribed to the glass transition of PVA [27]. In addition, an exothermic peak appeared at 296 °C, which was a little lower than that of pure PAN fiber. This may be due to the fact that the carboxyl groups of the PVA component in the PVA/PAN composite fiber are able to initiate the cyclization reaction by the ionic mechanism at a lower temperature [7]. The curve of C-PVA/PAN was similar to that of PVA/PAN, and the exothermic peak appeared at 309 °C, which is higher than that of the control sample (306 °C). The cross-linking between boric acid and the linear PVA chains, as discussed in Section 3.2, prevents the motion of the PVA chains, increases the quasi-crystalline structure and reduces the free volume in the amorphous regions of PVA [28]. Therefore, the collision probability of the carbonyl of PVA chains and the cyano groups of PAN chains decreases, at the same time, the addition of PVA hinders the contact between cyano groups. As a result, the cyclization reaction of cyano groups becomes difficult to some extent, resulting in an increase in the temperature of cyclization. Different from the mentioned fibers, FR-PVA/PAN showed an endothermic peak and two exothermic peaks. The endothermic peak was located at 101 °C, which was similar to the explanation of PVA/PAN. The exothermic peak at 172 °C was mainly contributed to the cold crystallization of PVA. In addition, the cyclization exothermic peak shifted to a lower temperature compared with the original PAN fiber. An explanation for this may be that the introduction of phosphorus-containing groups was able to catalyze the cyclization of PAN and accelerate the formation of the carbon layer due to the generated phosphorus-containing acids.

### 3.5. Combustion Properties

The LOI test is an easy and effective way to access the flame retardance of materials. The LOI values of PAN and FR-PVA/PAN fabric before and after different washing cycles are listed in Table 4. It is known that PAN fiber is a kind of flammable synthetic fiber with a LOI value of ca. 17%. As shown in Table 4, the LOI value of PVA/PAN composite fiber decreases a little, attributed to the flammability of the PVA component in the composite fiber. After cross-linking, the LOI value of C-PVA/PAN increases because boron is an effective flame-retardant element for PVA. For FR-PVA/PAN, the LOI value rises to 34.3%, showing an excellent flame-retardant performance, which is consistent with the TG analysis. With the number of washing cycles increasing, the LOI values of FR-PVA/PAN decrease a little. This indicates that FR-PVA/PAN has an excellent durable flame-retardant performance owing to the existence of the covalent bond of P–O–C between phosphorus-containing groups and PVA molecule chains, as is evidenced by FTIR.

A cone calorimeter (CC) is an effective bench-scale instrument to simulate the fire hazards of the material in a real fire. The time to ignition (TTI), peak of heat release rate (PHRR), time to peak heat release rate, total heat release (THR), smoke production rate (SPR), total smoke production (TSP), and fire growth rate index (FIGRA) of the control PAN and the FR-PVA/PAN are obtained and shown in Figure 5. These important parameters are summarized in Table 5. As shown in Table 5, the TTI increases from 25 s of PAN fiber to 33 s of FR-PVA/PAN, indicating the delayed ignition. Heat release properties are of great significance in assessing the flame-retardant properties of materials. Compared with the original PAN fiber, the PHRR of FR-PVA/PAN decreases from 374.4 kW/m^2^ to 149.0 kW/m^2^ with a 60.2% reduction, while the time to PHRR extends from 45 s to 55 s with a 22.2% extension. The THR decreases from 7.3 MJ/m^2^ to 4.5 MJ/m^2^. These results indicate that the flame-retardant properties of the FR-PVA/PAN composite fiber are highly improved.

Furthermore, the PSPR and TSP of the FR-PVA/PAN composite fiber decreased by 83.3% and 73.3%, respectively, which indicates a reduction in fire hazards. The average mass loss rate (aMLR) of the FR-PVA/PAN composite fiber is reduced by half compared with that of the control PAN fiber, while the residual mass of the FR-PVA/PAN composite fiber after combustion evidently increases compared with the control sample, which confirms the results of the TG test. The fire growth rate index (FIGRA) was a deduced parameter which equals the ratio of PHRR and time to PHRR and reflects the maximal fire hazard when a material is under real burning conditions [29]. The FIGRA value of the FR-PVA/PAN fiber decreases by 67.4%, demonstrating the suppression for both the fire occurrence and the fire spread [11].

### 3.6. Mechanism Analysis

#### 3.6.1. TG-FTIR Analysis

TG-FTIR was used to investigate the evolved gaseous ingredients of FR-PVA/PAN fiber during thermal decomposition at different temperatures. The FTIR spectra of the volatile gases decomposed from FR-PVA/PAN fiber at different temperatures are shown in Figure 6. As shown in Figure 6, gaseous ingredients were generated ranging from 200 °C to 800 °C. The peaks at 3736 cm^−1^ and 3630 cm^−1^ appeared at 400 °C and were attributed to the –OH group of the released water [30]. These peaks become more intense as the temperature increases and reach a maximum at 800 °C. The peak at 2960 cm^−1^ is attributed to C–H derived from the aliphatic species. The peak at 2360 cm^−1^ is identified as a CO_2_ stretching vibration [10,31] and with increasing temperature, the peak intensity greatly increases, which indicates that CO_2_ is the main decomposed gaseous compound. The weak peak at 2108 cm^−1^ appeared above 400 °C, assigned to the stretching vibration of CO [32]. The weak peak appeared at 1526 cm^−1^ above 400 °C and was ascribed to the combination of the stretching vibration of C–N and bending vibration of NH_3_ [10]. The peaks at 668 cm^−1^ and 760 cm^−1^ are assigned to the CO_2_ and C–H bending vibration of HCN, respectively [31]. They are attributed to compounds containing aromatic rings [33]. It is clearly observed that an adsorption peak at 1250 cm^−1^ of P=O [29] appeared above 200 °C, and reached a maximum at 700 °C. The released phosphorus-containing species, such as PO· and PO_2_· can act as effective free-radical scavengers to quench the gas-phase chain propagation reactions during combustion [34]. In addition, they can trap H· and HO· radicals in the gas phase, resulting in improved flame retardancy [34]. Furthermore, the inflammable gases (CO_2_ and H_2_O) can not only dilute the flammable gases in the gas phase but also act as a barrier to isolate external oxygen into the combustion zone. As a result, the combustion is inhibited, which is conducive to the improvement of the flame retardancy.

#### 3.6.2. Py-GC/MS Analysis

To confirm the detailed pyrolysis products in the gas phase, a pyrolysis–gas chromatography/mass spectrometry (Py-GC/MS) test under an air atmosphere was performed. Herein, FR-PVA/PAN was pyrolyzed at 537 °C. The total ion chromatograms are presented in Figure 7, and the corresponding peaks and their possible assignments are listed in Table 6. It has been reported [35] that the main pyrolyzed products of PAN are of dimers, trimers, tetramers, and their derivatives. 

As shown in Table 6, the main pyrolyzed gaseous products of FR-PVA/PAN are CO_2_, nitrile derivatives, such as acetonitrile, 2-propenenitrile, propanenitrile, methacrylonitrile, and isobutyronitrile. In addition, many nitrogen-containing compounds, such as 1H-pyrrolo [2,3-b] pyridine, 1,8-naphthyridine, 6-cyanoquinoline, 3-aminoquinoline, 1H-pyrido [2,3-b] indole, 1,2-diaminonaphthalene, etc are produced. Furthermore, aromatic compounds, such as benzene, anthracene, phenanthrene, 5,12-diphenyl-dibenz [a,h] anthracene, benzo [b] triphenylene, indeno [1,2,3-cd] pyrene, dibenz [a,h] anthracene, benzo [ghi] perylene appeared in the gas-phase. These results illustrate that under thermal pyrolysis, aromatic compounds are produced, indicating the successful cyclization reaction of cyano groups in FR-PVA/PAN and the formation of trapezoidal structure. Therefore, the char residue increases accordingly, which is conducive to the improvement of the flame retardancy of FR-PVA/PAN. It is consistent with the condensed-phase flame-retardant mechanism. In conclusion, upon fast pyrolysis of the flame-retardant FR-PVA/PAN, the released inflammable gas, such as CO_2_, not only dilutes the hot atmosphere at the combustion surface but also cuts off the supply of oxygen, playing a flame-retardant role in gas phase [36].

## 4. Conclusions

An easy and eco-friendly method for preparing flame-retardant polyacrylonitrle fiber has been successfully developed. Compared with the control PAN fiber, the tensile strength of the FR-PVA/PAN composite fiber increases by 55%. TGA indicates that the char residue of the FR-PVA/PAN composite fiber at 800 °C is 62.6 wt%, showing excellent char-forming ability. In contrast, the PHRR, THR, PSPR, and TSP of FR-PVA/PAN decrease by 60.2%, 38.4%, 83.3%, and 73.3%, respectively. In addition, the FIGRA value of the FR-PVA/PAN fiber decreases by 67.4%. These results indicate that the flame retardancy of FR-PVA/PAN improves significantly due to the incorporation of phosphorus, nitrogen, and boron elements. TG-FTIR and Py-GC/MS tests indicate that FR-PVA/PAN has the condensed- and gas-phase flame-retardant mechanisms. The inflammable CO_2_ dilutes the combustible gases and reduces the effective combustion heat of the volatiles. In addition, the prepared graphitized char residue acts as a barrier to prevent external oxygen from entering the combustion area and the heat escaping from the combustion area. This work simultaneously improves the strength of the composite fiber and retains the inherent properties of PAN. It also demonstrates that this flame-retardant scenario is feasible for preparing phosphorus-containing and high-strength flame-retardant PAN composite fiber.

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
