# Peer review of "Polyvinyl Alcohol Reinforced Flame-Retardant Polyacrylonitrile Composite Fiber Prepared by Boric Acid Cross-Linking and Phosphorylation"

_materials, 2018, doi:10.3390/ma11122391_

Round 1
Reviewer 1 Report
I consider that the authors have presented and interesting work on the development of fire retarded fibres. I have noticed that the authors have substantially improved the manuscript and I consider it is worth to publish this paper. However, the authors should still correct some language errors along the text.
I also include some comments to be considered.
3.3. XPS analysis
Line 231
Furthermore, the phosphorus and boron contents are up to 7.70% and 13.18%, respectively, which is in favor of improvement of flame retardancy of FR-PVA/PAN.
The authors could include a comment indicating the load that they expected to add to the PVA/PAN system. It is not clear if the values obtained are above or below the expected ones.
3.4. Thermal Stability
Line 264
Similarly, C-PVA/PAN and FR-PVA/PAN both possessed four thermal degradation stages.
From the DTG curve, it seems that FR-PVA/PAN occurs in only 3 steps.
Line 277
This last paragraph introduces results that cannot be seen with the thermal analysis. I recommend removing it from here because they are already mentioned in the conclusions section.
3.5. Combustion properties
Line 316
With the washing cycles increasing, the LOI values of 316 FR-PVA/PAN decrease a little. This indicates that the flame retardant performance is well maintained
After
30 cycles, the LOI values decreased from 34.3 to 30.6. I think that the
authors should reformulate the paragraph to indicate what does it mean well maintained. It should be interesting to know the remaining amount of P and N in the washed samples.
Author Response
Dear reviewer,
First of all, thank you for your hard work for reviewing our paper. According to your comments, we revised the article carefully. The detailed revision is given below.
3.3. XPS analysis
Line 231
Furthermore, the phosphorus and boron contents are up to 7.70% and 13.18%, respectively, which is in favor of improvement of flame retardancy of FR-PVA/PAN.
The authors could include a comment indicating the load that they expected to add to the PVA/PAN system. It is not clear if the values obtained are above or below the expected ones.
Answer: We added the comment of phosphorus content in flame retardanting PAN in the XPS analysis section and marked them in red color. However, the boron content in flame retarding PAN has no literature.
3.4. Thermal Stability
Line 264
Similarly, C-PVA/PAN and FR-PVA/PAN both possessed four thermal degradation stages.
From the DTG curve, it seems that FR-PVA/PAN occurs in only 3 steps.
Line 277
This last paragraph introduces results that cannot be seen with the thermal analysis. I recommend removing it from here because they are already mentioned in the conclusions section.
Answer: We revised the description of C-PVA/PAN and FR-PVA/PAN. And we deleted the last sentence of the last paragraph.
3.5. Combustion properties
Line 316
With the washing cycles increasing, the LOI values of 316 FR-PVA/PAN decrease a little. This indicates that the flame retardant performance is well maintained
After 30 cycles, the LOI values decreased from 34.3 to 30.6. I think that the authors should reformulate the paragraph to indicate what does it mean well maintained. It should be interesting to know the remaining amount of P and N in the washed samples.
Answer: We revised the sentence as follows: With the washing cycles increasing, the LOI values of FR-PVA/PAN decrease a little. This indicates that FR-PVA/PAN has excellent durable flame retardant performance, resulting from the covalent bond of P-O-C between phosphorus-containing groups and PVA molecule chains, as is evidenced by FTIR.
Reviewer 2 Report
The comments and suggestions are given in the document attached. Please revise the title with the better use of English language.

Author Response
Dear reviewer,
First of all, thank you for your hard work for our paper. According to the comments, we revised the article carefully. The detailed revision is given below.
Introduction. This is a well-written section and introduces the topic very well. Some suggestions for improvement include:
1a) line 37 ‘wonderful’ consider a synonym;
Answer: We changed wonderful into good.
1b) line 38 replace ‘limit’ with ‘limiting’;
Answer: Limit was replaced with limiting.
1c) line 44 use ‘macromolecular’ rather than ‘molecule’;
Answer: We changed molecule with macromolecular.
1d) lines 45-47, I disagree with a statement mentioned a ‘relatively complex synthesis’ as the authors of the references [1,3] stress the relative simplicity on the synthetic procedure. Please revise this sentence;
Answer: We deleted “relatively complex synthesis” and changed the sentence of “In this scenario, those monomers containing flame retardant elements, especially phosphorus-containing comonomers must be previously undergone molecule design and relatively complex synthesis. On the one hand, the cost will arise to some extent, on the other hand, the comonomers” with “In this scenario, those monomers containing flame retardant elements, especially phosphorus-containing comonomers must be previously undergone molecule design, in addition, the cost will arise to some extent, and the comonomers”
1e) line 48 replace ‘extend’ with ‘extent’, check the reminder of the text for this error as it is repeated later;
Answer: We replaced “to some extend” with “to some extent”.
1f) line 49, use ‘molecular weight’ rather than ‘molecule weight’, check the reminder of the text for this error as it is repeated later;
Answer: We replaced ‘molecule weight’ with ‘molecular weight’.
1g) lines 60 and 90, correct ‘dydrazine’, I suspect it should be ‘hydrazine’;please correct;
Answer: Dydrazine was replaced with hydrazine.
1h) line 72, in the name of the chemical KH550 you should write ‘amino’ instead of ‘amina’, please correct;
Answer: We 3-aminapropyltriethoxysilane with 3-aminopropyltriethoxysilane.
1i) line 75, take out ‘all’ from the sentence;
Answer: We deleted “all” from the sentence.
1j) line 82 please specify an average range of how much of flame retardants needs to be added;
Answer: The addition of flame retardants is over 25 wt%.
1k) line 83, please specify what mechanical properties of the fibres will decrease;
Answer: We replaced “mechanical properties” with “breaking strength”.
1l) line 86, please indicate what amount of PVA was blended as per reference [14]; Answer: The amount of PVA is 30-50 wt%.
1m) lines 97 and 97 please indicate exactly what is the concentration of boric acid instead of a general term ‘a certain concentration’.
Answer: The concentration of boric is ca. 5 wt%.
1. Experimental section.
Please describe the process of preparing the control samples of PAN and PVA/PAN fibres in the individual sub-sections.
Answer: I think if the control sample of PAN and the PVA/PAN fibers are described in the individual sub-section, it appears somewhat wordy.
2a) line 111, replace ‘bound’ with ‘round-bottom’;
Answer: “bound” was replaced with “round-bottom”.
2b) line 114, consider a different term for ‘debubbled’;
Answer: “debubbled” was replaced with “deaerated”.
2c) line 116, please state the concentration of boric acid and indicate the solvent in the coagulation bath. Is it water?
Answer: The concentration of boric acid aqueous solution is ca. 5 wt%. Yes, it is water.
2d) lines 118-119, I am wondering if drying in a vacuum oven at 60oC for 6 hrs is sufficient time for drying? The following TGA showed that the samples begin to lose mass early (Figure 3), meaning that the moisture was not eliminated at the drying stage’;
Answer: It really dries completely under the condition because the fiber is finer. However, due to the amounts of hydroxyl groups in the composite fiber, it will adsorb moistrue, as a result, when it is heated, the adsorbed water will release, resulting the lower decomposition temperature.
2e) lines 119-120, it is not clear what crosslinking the authors are referring to, ‘thermal’ or ‘boric acid’. I suggest to revised this paragraph to indicate clearly that boric acid acts as a cross-linking agent, is this true? I also assume that PAN fibres did not contain PVA, is this correct? The authors also mention in the line 117 that cross-linked PVA/PAN fibres are made through coagulation in the hot water bath – it is unclear what fibres are cross-linked, those that are labelled as PVA/PAN or C-PVA/PAN. Please clarify this in the relevant sub-section.
Answer: The PVA/PAN composite fiber was cross-linked by boric acid. The sentence was revised as “The spinning dope was extruded from a spinneret orifice (0.05 mm) into the coagulation bath (80 oC) of boric acid aqueous solution with the concentration of ca. 5 wt% to cross-link.” In fact, in the work, PAN fiber is fabricated by pure PAN, while PVA/PAN composite fiber is obtained from PAN and PVA blended polymers.
2f) line 116, please indicate the size of the orifice and the equipment used to make the fibres;
Answer: The orifice size is 0.05 mm and the equipment used was designed and manufactured by ourselves.
2g) line 123, please indicate the amount (weight in g or %) of C-PVA/PAN used;
Answer: 1 g of C-PVA/PAN was used.
2h) Scheme 1 shows that only OH groups of PVA are esterified by phosphoric acid. I am wondering if CN groups underwent any transformations under the effect of phosphoric or boric acids?
Answer: We revised Scheme 1. In fact, boric acid has little influence on PVA/PAN composite fiber because of weak acidity. Compared with boric acid, phosphoric acid has stronger acidity, but the time of treatment is not long and the treat temperature is not high. Therefore, although the cyanide can hydrolyze and form carboxyl group, but the amount of carboxyl groups is not much.
2i) The authors carried out various characterisations on the samples of individual fibres and woven fabrics. It should be clearly stated for each characterisation technique used in this study what form of the sample is used. For example, FT-IR was recorded on disks, films, powder?
Answer: In this article, fabric state was used to test for different properties.
2j) The XPS instrument details are not given;
Answer: XPS instrument details were given.
2k) Please include the size of the samples for cone calorimeter; also explain here what is FIGRA;
Answer: FIGRA (the ratio of PHRR and time to PHRR) was defined.
2l) The country of instruments manufacturers, lines 165 and 172, should be USA rather than American;
Answer: We changed American with USA.
2m) line 169, please revise the sentence as Py-GC/MS provides information about the nature of gaseous products of pyrolysis, not about the molecular structures of FC-PVA/PAN fabric;
Answer: We have deleted “the molecular structrues”, and changed with gaseous products.
2n) line 164, take out ‘Is’;
Answer: “Is” is deleted.
2. Results and discussion.
3a) line 176, please use ‘properties’ instead of ‘property’;
Answer: We have replaced property with properties.
3b) revise the third sentence of the first paragraph of sub-section 3.1. It is very difficult to understand the explanation given from such a long sentence;
Answer: The revised sentence is as follows: It is generally known that PVA and PAN are two different macromolecules. They have different crystalline strucure and motion ability. In addition, inter and intra hydrogen bonds between PVA and PAN macromolecules restrain the movement of the two macromolecules. Thus, under the same spinning conditions, the two macromolecular chains may be crimp and not drawn completely during stretching.
3c) line 183; use ‘macromolecular’ instead of ‘molecular’; take out the word ‘molecule’; there are three words derived from ‘drawing’ in the short part of the sentence – please re-phrase;
Answer: We have deleted molecular and replaced with macromolecular. At the same time, we deleted molecule. We replaced drawing with stretching.
3d) I tend to disagree with the statement that ‘the cross-linking reaction between boric acid and PVA does not change the molecular orientation and the arrangement’ as per lines 185-186 – the crosslinking changes the arrangement of the polymeric chains by generating a network of bonds. Please correct;
Answer: The comment of the reviewer is right. Therefore, we revised the sentence as follows: the cross-linking reaction between boric acid and PVA changes the macromolecular orientation and the arrangement by generating a network.
3e) line 187 correct’phohphorylation’; revise this sentence by removing its last part (‘, resulting …’);
Answer: The word “phohphorylation” was wrongly written, and was changed to “phosphorylation”. And the sentence was revised with “PVA/PAN composite fiber and the modified composite fiber are all increased.”
3f) lines 189-190, I disagree that ‘tensile strength of the samples, PVA/PAN composite fiber and the modified composite fiber are all increased greatly’, this is just an increase, not a great increase; what is ‘excellent enhancing ability of PVA’ – please explain and revise;
Answer: “greatly” was deleted. We changed the sentence with “The increased tensile strength of PVA/PAN composite fiber and the modified composite fiber may be due to the presence of some interaction between PAN and PVA macromolecules in the blend”.
3g) There is no discussion about the breaking elongation parameter, which is reported in Table 1 but not discussed in the text. Please explain why the breaking elongation is decreased in the modified fibres;
Answer: The analysis of breaking elongation was added in the text.
3h) line 199, please explain what ester group is responsible for C=O vibration signal; Answer: It is vinyl acetate.
3i) lines 202-203, the peak at 1630 cm-1 was assigned to ‘acetate groups remaining in partially hydrolysed PVA’. However, this peak was not observed in the spectra of PAN, which is, according to the authors description in 2.1 sub-section, contained 5 wt% of vinyl acetate. Explain why;
Answer: In fact, there is a weak peak at 1630 cm-1 in the PAN FTIR spectrum, but when the spectra of all the samples are drawn in one figure, the intensity of the peak at 1630 cm-1 in the FTIR spectrum of PAN becomes weaker and almost invisible. The FTIR spectrum of PAN fiber is listed below.
3j) lines 203-205, the authors claimed that ‘For C-PVA/PAN composite fiber, new absorption peaks could be clearly observed. These absorption peaks located at 1450 cm-1, 1230 cm-1, 772 cm-1 and 653 cm-1 were assigned to the stretching vibration of B-O-C, and O-B-O’. I would tend to disagree that peaks 1450 cm-1 and 1230 cm-1are new since the authors just a few lines earlier (see line 198-200) assigned them to other groups. Please clarify and correct;
Answer: We changed the sentence of “For C-PVA/PAN composite fiber, new absorption peaks could be clearly observed. These absorption peaks located at 1450 cm-1, 1230 cm-1, 772 cm-1 and 653 cm-1 were assigned to the stretching vibration of B-O-C, and O-B-O” into “For C-PVA/PAN composite fiber, new absorption peaks located at 772 cm-1 and 653 cm-1 were assigned to the stretching vibration of B-O-C, and O-B-O”.
3k) lines 207-209, the authors state that ‘the peak at 1650 cm-1 was attributed to the carboxyl group (- COOH), which demonstrated that the CN in PAN was hydrolyzed continuously in the phosphoric acid solution’ – please show the result of this conversion on Scheme 1;
Answer: Scheme 1 has revised and shown the conversion.
3l) line 211, it is a very strong statement that ‘These results verified the successful chemical reaction’ – the FT-IR can be used in conjunction with other methods such as XPS or NMR to confirm the chemical transformations of the functional groups. Please revise;
Answer: We delete the sentence “These results verified the successful chemical reaction between C-PVA/PAN and phosphoric acid.”
3m) please label the vertical axis on the graphs depicted in Figures 1, 2 and 6; Answer: The three figures have been revised.
3n) in sub-section 3.3 please confirm what the percentages you are referring to, I assume wt. %?; also, please explain why there is no B signal for C-PVA/PAN – I thought as per your earlier Scheme 1 it should contain B element;
Answer: the percnetage of elements is mole percentage. In fact, B signal is present in C-PVA/PAN spectrum, because a lot of curves are drawn in a picture, the peak appears smaller, as is similar with FTIR spectra.
3o) line 219, please revise ‘significantly’ – incorrect use;
Answer: We deleted significantly and revised as “showed an increase in the content of oxygen element”.
3p) I am wondering whether the authors have carried TGA in nitrogen atmosphere and if ‘yes’ it would be useful to present these results here; they will correlate better with DSC measurements;
Answer: In order to access the real thermal oxidation decomposition of the samples, we performed TGA in air instead of in nitrogen.
It would be useful to include in the text the values of char residues obtained in TGA experiments for all the samples not just for FR-PVA/PAN.
Answer: The char residues of other samples are given in the text.
The detailed description of mass losses was given for PAN, PVA/PAN and C-PVA/PAN. However, the same level of detail is missing for FR-PVA/PAN: only the amount of char produced is stated. Please include this information in sub-section 3.4.
Answer: We deleted the sentence “And the two fibers possessed the same thermal degradation trend”. However, If we narrate the decomposition of FR-PVA/PAN as that of C-PVA/PAN again, it will be wordy. Therefore, we focus on the analysis of the reasons for the increase of char residue of FR-PVA/PAN.
3q) lines 266-267, I would disagree with the authors in the statement that ‘FR-PVA/PAN and C-PVA/PAN showed a similar decompositon trend’. As it follows from Figure 3a, the sample of fire retardant fibre has a TG profile which differs from other samples. Please correct;
Answer: We deleted “FR-PVA/PAN and C-PVA/PAN showed a similar decompositon trend’.
3r) lines 268-269, it is worthwhile mentioning the role of phosphoric acid species in formation more stable char residue structure. Please use the information available in the recent publication in Polymers 2018, 10, 131; doi:10.3390/polym10020131.
Answer: Sorry, we do not find the article through “doi:10.3390/polym10020131.”
3s) On figure 1b the label B-PVA/PAN is not explained. Please correct.
Answer: It is mistake writing and revised.
3t) line 279, use ‘shoulder’ instead of ‘acromion’;
Answer: acromion was changed to shoulder.
3u) line 283, please correct ‘carbony’;
Answer: It is mistake writing and revised.
3v) line 308, change ‘flammable’ to ‘flammability’;
Answer: ‘flammability’ was changed to ‘flammable’.
3w) lines 311-313, please revise this sentence; it does not make sense in the current from;
Answer: With the washing cycles increasing, the LOI values of FR-PVA/PAN decrease a little. This indicates that the flame retardant performance is well maintained, which is owing to the covalent bond of P-O-C between phosphorus-containing groups and PVA molecule chains, as is evidenced by FTIR.
3x) line 321, please replace ‘suppressed burning rate’ with ‘delayed ignition’;
Answer: We replaced ‘suppressed burning rate’ with ‘delayed ignition’
3y) line 328, explain abbreviation aMLR, not clear as this parameter is not given in the table 5;
Answer: aMLR was given the full name.
3z) Table 5, please provide units for FIGRA column.
Answer: We added the unit of FIGRA in Table 5.
3. Conclusions
Please revise the first sentence as the large scale production was not attempted in the study. Perhaps, this method could be suitable for the large scale production. The authors should focus their conclusions on the final fire-retardant product, FR-PVA/PAN. Thus, in line 394 correct ‘PVA/PAN’ to ‘FR-PVA/PAN’. Please make the conclusion about mechanical properties of the final fibre. I would take out the statement about the poor thermal stability of the fibres and stress on the excellent flame retardancy achieved. Also, remember to include the role of phosphoric acid species in the improved ability to form stable chars.
Answer: We deleted large scale production. The revised sentence is as: Compared with the control PAN fiber, blended PVA increases the tensile strength of FR-PVA/PAN composite fiber with ca. 55% increasment. At the same time, we changed “The result of TGA revealed PVA/PAN and the modified composite fibers were less thermally stable than pristine PAN fiber.” into “TGA indicated that the char residue of FR-PVA/PAN composite fiber at 800 oC was 62.6% showing excellent char-forming ability”
Reviewer 3 Report
The authors have tried to improve the strength while maintaining the inherent properties of flame retardant with polyacrylonitrile (FR-PAN) fibers. The paper is well written and nicely presented.
However, there are some comments to improve the manuscript.
1. The authors should rewrite the manuscript since there were some paras found to be plagiarized, even from same authors. " Yue Zhang, Yuanlin Ren, Xiaohui Liu, Tongguo Huo, Yiwen Qin. "Preparation of durable flame retardant PAN fabrics based on amidoximation and phosphorylation", Applied Surface Science 2018". The similarity is obvious, starting from line 152 to 174.
2. Please improve the English language of the manuscript, even it takes professional assistance.
3. The abstract of the conclusion should be in same consent, the conclusion is superficial, as it states the large scale production of PVA/PAN and "this work is the 404 first report on simultaneously improving the strength and retaining the inherent properties of 405 durable flame retardant of PAN fiber. And the present work demonstrated that this flame retardant 406 scenario was feasible for preparing eco-friendly and high strength flame retardat PAN fiber".
But the different papers can be found in literature, most importantly or most recently the same manuscript mentioned above, try to report similar approach.
4. With the incorporation of FR into the PVA/Pan fibers, the mechanical properties seems to improve and the elongation is decreased. It should be explained in detail.
Author Response
Dear reviewer,
First of all, thank you for your hard work for our paper. According to the comments, we revised the article carefully. The detailed revision is given below.
1. The authors should rewrite the manuscript since there were some paras found to be plagiarized, even from same authors. " Yue Zhang, Yuanlin Ren, Xiaohui Liu, Tongguo Huo, Yiwen Qin. "Preparation of durable flame retardant PAN fabrics based on amidoximation and phosphorylation", Applied Surface Science 2018". The similarity is obvious, starting from line 152 to 174.
Answer: We have revised them accoring to the reviewer’s comments.
2. Please improve the English language of the manuscript, even it takes professional assistance.
Answer: We have checked the English language of the manuscript carefully.
3. The abstract of the conclusion should be in same consent, the conclusion is superficial, as it states the large scale production of PVA/PAN and "this work is the 404 first report on simultaneously improving the strength and retaining the inherent properties of 405 durable flame retardant of PAN fiber. And the present work demonstrated that this flame retardant 406 scenario was feasible for preparing eco-friendly and high strength flame retardat PAN fiber".
But the different papers can be found in literature, most importantly or most recently the same manuscript mentioned above, try to report similar approach.
Answer: The last statement of the conclusion was revised as follow: This work simultaneously improves the strength of the composite fiber and retains the inherent properties of PAN. And it demonstrates that this flame retardant scenario is feasible for preparing phosphorus-containing and high strength flame retardat PAN composite fiber.
4. With the incorporation of FR into the PVA/Pan fibers, the mechanical properties seems to improve and the elongation is decreased. It should be explained in detail.
Answer: The elongation was analyzed in the text.
Round 2
Reviewer 2 Report
The authors have addressed most of the comments and corrected the manuscript accordingly.
In addition to correcting the English language, there are some minor changes needed:
As the authors indicate the condensed phase mechanism of fire retardance they must include discussion about the actions of phosphoric acid derivatives on char surfaces. The authors were referred to an open access publication where this role was reported. The publication is available from https://www.mdpi.com/2073-4360/10/2/131. Alternatively, the confirmation of condensed phase mechanism of fire retardance should derive from solid state NMR of char residues obtained in the study.
Please improve style of conclusion statements by rephrasing poorly constructed sentences.
Author Response
Dear reviewer,
First of all, thank you for your hard work for reviewing our paper again. According to your comments, we revised our article carefully and the revision is given below.
1) In addition to correcting the English language, there are some minor changes needed:
As the authors indicate the condensed phase mechanism of fire retardance they must include discussion about the actions of phosphoric acid derivatives on char surfaces. The authors were referred to an open access publication where this role was reported. The publication is available from https://www.mdpi.com/2073-4360/10/2/131. Alternatively, the confirmation of condensed phase mechanism of fire retardance should derive from solid state NMR of char residues obtained in the study.
Please improve style of conclusion statements by rephrasing poorly constructed sentences.
Answer: We revised the grammar again. And, the action of phosphoric acid species on char-forming was also discussed and reffered to the reference of 24. The rest of the references’ sequence has been adjusted. At the same time, the conclusion statements were also revised.
Reviewer 3 Report
The authors have revised the manuscript according to the suggestions.
Author Response
Dear reviewer,
Thank you again for reviewing our paper. According to your comments, we do not need to revise our article.
1) The authors have revised the manuscript according to the suggestions.
Answer: We agree with the comments of the reviewer.